# Comparing Seamounts and Coral Reefs with eDNA and BRUVS Reveals Oases and Refuges on Shallow Seamounts

**DOI:** 10.3390/biology12111446

**Published:** 2023-11-17

**Authors:** Florian Baletaud, Gaël Lecellier, Antoine Gilbert, Laëtitia Mathon, Jean-Marie Côme, Tony Dejean, Mahé Dumas, Sylvie Fiat, Laurent Vigliola

**Affiliations:** 1ENTROPIE, Institut de Recherche pour le Développement (IRD), UR, UNC, IFREMER, CNRS, Centre IRD de Nouméa, 98848 Noumea, New Caledonia, France; florian.baletaud@soproner.nc (F.B.); gael.lecellier@unc.nc (G.L.); laetitia.mathon@gmail.com (L.M.); mahe.dumas@ird.fr (M.D.); sylvie.fiat@ird.fr (S.F.); 2GINGER SOPRONER, 98000 Noumea, New Caledonia, France; antoine.gilbert@soproner.nc; 3GINGER BURGEAP, 69000 Lyon, France; jm.come@groupeginger.com; 4MARBEC, University of Montpellier, CNRS, IFREMER, 34000 Montpellier, France; 5ISEA, University of New Caledonia, 98800 Noumea, New Caledonia, France; 6CEFE, University of Montpellier, CNRS, EPHE-PSL, IRD, 34000 Montpellier, France; 7SPYGEN, 73370 Le Bourget-du-lac, France; tony.dejean@spygen.com

**Keywords:** conservation, biomass, biodiversity, hotspot, coral reefs, mesophotic slope

## Abstract

**Simple Summary:**

Underwater mountains, or seamounts, are deep-sea habitats collectively forming an area as large as Europe. Yet, they are one of the least studied ecosystems on earth. Known for supporting rich marine life compared to surrounding deep-sea environments, we have no information on how seamounts truly compare to other iconic biodiversity hotspots like shallow coral reefs. To assess the effective ecological value of seamounts, we compared fish communities in coral reefs and seamounts up to 500 m deep using two techniques: environmental DNA to detect the presence of species by filtering fragments of DNA lost by organisms in seawater, and underwater cameras to directly measure fish abundance and size. We found that the deepest seamounts had almost 10 times fewer fish species than coral reefs. However, the shallowest seamounts had larger fish species, including sharks, than coral reefs. We conclude that while seamounts are important and unique ecosystems, they may not be as diverse for fish species as previously thought (diversity hotspots) but rather biomass oases and refuges for endangered species. This study therefore calls for protecting the shallowest seamounts, as they are critical areas for marine life.

**Abstract:**

Seamounts are the least known ocean biome. Considered biodiversity hotspots, biomass oases, and refuges for megafauna, large gaps exist in their real diversity relative to other ecosystems like coral reefs. Using environmental DNA metabarcoding (eDNA) and baited video (BRUVS), we compared fish assemblages across five environments of different depths: coral reefs (15 m), shallow seamounts (50 m), continental slopes (150 m), intermediate seamounts (250 m), and deep seamounts (500 m). We modeled assemblages using 12 environmental variables and found depth to be the main driver of fish diversity and biomass, although other variables like human accessibility were important. Boosted Regression Trees (BRT) revealed a strong negative effect of depth on species richness, segregating coral reefs from deep-sea environments. Surprisingly, BRT showed a hump-shaped effect of depth on fish biomass, with significantly lower biomass on coral reefs than in shallowest deep-sea environments. Biomass of large predators like sharks was three times higher on shallow seamounts (50 m) than on coral reefs. The five studied environments showed quite distinct assemblages. However, species shared between coral reefs and deeper-sea environments were dominated by highly mobile large predators. Our results suggest that seamounts are no diversity hotspots for fish. However, we show that shallower seamounts form biomass oases and refuges for threatened megafauna, suggesting that priority should be given to their protection.

## 1. Introduction

Seamounts are ubiquitous deep-sea habitats that collectively form an area as large as Europe [1]. Although seamounts are often considered biodiversity hotspots [2], oases [3,4], and refuges for marine megafauna [5], they are the least studied landforms of the major biomes of the global ocean. Out of more than 170,000 seamounts around the world, only less than 0.002% have been sampled for scientific purposes [6]. Yet, evidence is accumulating that seamounts are increasingly threatened by anthropogenic impacts such as overfishing, destructive fishing, and marine mining [6,7]. Thus, it is essential to fill the large gaps in our knowledge regarding the diversity and abundance patterns of species on seamounts. This is especially urgent at a time when human-related disturbances are affecting all areas of the world and oceans may experience a mass extinction of sufficient intensity to rank among the major extinctions of the Phanerozoic (541 Ma to present), with vertebrates, including fish, being at the forefront [8].

Seamounts are generally defined as underwater mountains greater than 100 m in relief above the seafloor, with often further subdivision into hills (elevation < 500 m from seafloor), knolls (>500 m from seafloor), and seamounts (>1000 m from seafloor) [9]. In geology, oceanic islands are also considered emerging seamounts [10]. Some ecologists also regard remote atolls as seamounts [11]. Above all, seamounts are geological features that can modify the properties of their surrounding oceanic environment. In particular, some studies indicate higher biodiversity and biomass on seamounts than in surrounding abyssal and pelagic environments [12,13]. Although many factors, including physical and biogenic habitat structure, may explain such patterns, the complex effects of seamounts on ocean circulation, including Taylor column formation, tidal amplification, internal waves, and upwelling formation, are thought to enhance primary production, zooplankton abundance, attract pelagic predators above seamounts, and permit these deep-sea habitats to host large populations of demersal and benthopelagic fish [6,14,15].

Due to their higher biodiversity than their surrounding environment, seamounts have long been considered hotspots, although this claim is now widely debated [16]. The term “biodiversity hotspot” was first introduced in 2000 and defined by delineating subjective regions of high (>1500 species, or 0.5% of the world’s richness) endemic vascular plant richness under threat [17]. This analysis was supported by vertebrates’ data but clearly warned against different congruence levels depending on the region. The marine realm also adopted the hotspot terminology and used the number of species, proxies of endemism, and global threats [18,19]. Today, “hotspot” has become a broader term that encompasses more biodiversity traits such as functional, beta, and phylogenetic diversity, leading to different definitions and uses. For this study, we consider the original definition of a marine hotspot based on biodiversity, which includes species richness and its previously stated derivatives. However, the identification of hotspots still relies on a comparison with other ecosystems, implying that biodiversity on seamounts should be compared to other recognized hotspots such as coral reefs (i.e., the shallow tropical coral reefs formed by reef-building corals associated with mostly photosynthetic coral assemblages), rather than merely surrounding abyssal and pelagic environments.

Biomass is another index that can refer either to abundance or mass, which directly connects to ecosystem services, for instance, through resources [20]. Temperature, nutrient availability, human influence, and biodiversity (where evolutionary adaptation results in species-rich communities using a larger fraction of available resources) are among the main factors controlling biomass production, at least in marine fish [21]. The concept of an ecosystem with higher biomass than its surroundings typically refers to an “oasis”. Originally defining a fertile spot in a desert due to abundant water [22], attempts were made to translate the term to the marine realm with hydrothermal vents in the abyss, as they basically represent an isolated ecosystem lying within a desert, but where previously limiting factors allow a drastic increase in biomass [23]. This definition has further been used to praise coral reefs [24], but also seamounts [3,4] and other deep-sea landforms, e.g., canyons [25]. However, definitions of marine oases and hotspots often mix and shift towards diversity oasis [26], a synonym of the hotspot, or biomass hotspot, a synonym of oasis. Here, we consider the term oasis as the original analogy, which implies that biomass levels are higher in the studied environment (e.g., seamounts) than in their surroundings [3,23].

The definition of an oasis does not have particular regard for anthropogenic threats, but the degree of isolation of an oasis may straddle the line with the concept of “refuge”. The original notion of refuge implies the existence of safe havens for species impacted by large-scale disturbances like the effects of climate change [27,28,29]. This definition has recently been enhanced to accept a smaller scale of direct threat avoidance for mobile species like fish [30,31]. Particularly, the “depth refuge” hypothesis implies that species may use deeper, ecologically less favorable environments at the extreme of their ecological niche as an avoidance pathway to short-time-scaled disturbances like fishing [32] or climatic events like storms [29,33]. Refuges include habitats such as seamounts [5] and remote coral reefs [34,35], where low accessibility to fishing fleets can create an economic barrier to harvesting [36] and induce a high abundance of shallow-water predators such as reef sharks that are otherwise largely extirpated [37].

Considering their wide definition, seamounts include a vast range of environmental conditions, from the warm photic zone at the surface of oceans to the cold aphotic zone on abyssal seafloors, and therefore show high variation in their biological communities, e.g., [38]. The question is thus no longer to know if seamounts are hotspots, oases, or refuges, but to assess which environmental conditions enhance biodiversity, biomass, and threatened species abundance across a wide environmental and human impact gradient and compare various seamount environments with well-studied and recognized hotspots such as coral reefs.

New Caledonia is a vast (1.4 million km^2^ economic exclusive zone—EEZ) South Pacific archipelago composed of a main island, large surrounding islands (Loyalty Islands), and many islets, approximately 1200 km east of Australia in the Coral Sea [39]. The archipelago has one of the largest barrier reefs in the world, covering 24,000 km^2^, and hosts one-third of the world’s most remote and wilderness reefs [36]. Its extraordinarily rich shallow coral reefs [40] were added to the UNESCO World Heritage List in 2008. The archipelago also includes 80 hills, knolls, and seamounts, with at least 19 larger seamounts >1000 m in height from the seabed (hereafter seamounts), whose summit depths range from near surface at 4 m to 2400 m [41]. Thus, New Caledonia is probably the ideal site to compare seamount biodiversity and biomass with those of another iconic biodiversity hotspot, biomass oasis, and megafauna refuge such as coral reefs.

Comparing biodiversity between shallow and deep environments is challenging due to the use of specialized sampling methods. For example, coral reefs are typically surveyed by divers [31], while seamount fauna is generally surveyed by experimental fishing, acoustic echosounders, and ROVs. However, new technologies such as environmental DNA (eDNA) metabarcoding and video surveys allow the collection of quantitative data in a standardized way in almost all marine ecosystems. The metabarcoding of eDNA is based on the retrieval and analysis of genetic material naturally released by organisms in their environments. It was recently shown to outperform dive and video surveys for estimating marine biodiversity [37,42], with a higher capacity to detect small, cryptic, low-density, and elusive species [43,44,45,46]. Yet, the drawback of eDNA metabarcoding is the lack of knowledge about organism size and biomass. Stereo Baited Remote Underwater Video Stations (BRUVS) can efficiently estimate species abundance and biomass in virtually any marine habitat [47,48], so the two methods seem complementary to compare biodiversity across a long depth gradient.

In this study, we collected eDNA and BRUVS data from seven shallow coral reefs, four shallow seamounts, four deep continental slopes, three deep seamounts, and four seamounts of intermediate depth in the New Caledonian EEZ. Taking advantage of this unique dataset, we modeled the effect of key environmental and human variables on fish biodiversity, abundance, and biomass using boosted regression trees (BRT) [49]. Then we compared fish fauna on seamounts of variable summit depth, deep continental slopes, and shallow coral reefs and evaluated if and when seamounts qualify as biodiversity hotspots, biomass oases, and refuges for megafauna.

## 2. Materials and Methods

### 2.1. Data Collection

Data was collected during four oceanographic campaigns aboard the R/V Alis in April and June 2019 and August and September 2020, and during six coastal trips from September to December 2019. We sampled 22 sites, including seven barrier coral reefs, four deep continental slopes along the west coast of Grande Terre, and 11 seamounts (>1000 m in height from the seabed) summits of variable depth (45–511 m) across the New-Caledonian archipelago (Figure 1, Appendix A, Appendix A). All samples were collected at the bottom. The seamounts were chosen to have different summit depths corresponding to euphotic, intermediate, and aphotic zones: four seamounts had summits higher than 200 m depth, four seamounts had summits between 200 and 320 m depth, and three seamounts had summits between 320 and 500 m depth. Coral reefs were sampled between 2 and 28 m depth, virtually covering their full depth range, continental slopes between 80 and 235 m, and seamounts between 45 and 570 m. Altogether, 224 BRUVS were deployed (5–16 per site) and 192 eDNA samples (6–10 per site) were collected in five environmental strata: coral reefs (average sample depth 13 m, SD ± 7 m), shallow seamounts (average 60 m, SD ± 8 m) labeled “Seamount (50 m)”, continental slopes (average 142 m, SD ± 45 m) labeled “Continental slope (150 m)”, intermediate seamounts (average 265 m, SD ± 36 m) labeled “Seamount (250 m)” and deep seamounts (average 498 m, SD ± 33 m) labeled “Seamount (500 m)”.

### 2.2. Stereo Baited Remote Underwater Video Stations (BRUVS)

Sampling started in the daylight morning. Each BRUVS was deployed 300–500 m apart on the coral reef and up to 1 km on seamount summits to prevent fish individuals from appearing on multiple videos and to ensure the independence of samples [48]. A BRUVS rig consisted of a protective metal structure holding two horizontally aligned cameras facing a bait canister attached at the end of a 1.5 m bar [51]. For each deployment, one kilogram of crushed sardines (*Sardinops* spp.) filled the canister. BRUVS were weighted and attached to a rope leading to a surface buoy. The stereo pair of cameras were separated by 800 mm, with a convergent angle of 8°. GoPro Hero 4 cameras were used and set to a medium field of view (FOV) in 1920 × 1080-pixel format, running at 30 frames per second.

Soaking times were calculated from the time BRUVS reached the seabed (t_0_) to t_0_ + 60 min. Fish were visually identified and counted on video using the EventMeasure software (version 5.42 (64 bit), released April 2020, www.seagis.com.au). We used the MaxN method (corresponding to the maximum number of fish for each species counted in an image across the video), which is until now the standard method [47,48,52].

Stereo measurement was made available with the recording of three claps before deployment to synchronize frames. Calibration was done using CAL software (version 3.25 (64 bit), released March 2019, www.seagis.com.au). Fork length (FL) of individual fish was measured, when possible, up to a limit of 10 individuals per BRUVS per species to optimize video processing time.

### 2.3. Biomass Estimation

Biomass was calculated for each species on each BRUVS using the length-weight relationship$\;Weight\;\left(g\right)=a\times {Length\;\left(cm\right)}^{b}$, with *a* and *b* [53] retrieved from FishBase (https://www.fishbase.se (accessed on 12 February 2023)), and fish length calculated as the average length of all measured individuals (up to 10) of a species in a BRUVS [53]. When particular species could not be measured on a single BRUVS, the missing species length was estimated by data imputation using the MissForest algorithm with 999 trees [54]. We imputed the missing length using measured length records of other samples, but also family, genus, maximum size, and size type from Fishbase. The latitude and longitude of the localities where length records were taken were also used to account for the geographic proximity of measured lengths. The MissForest accuracy was tested with a k-fold cross-validation procedure by predicting 5% of the lengths each time by training the missForest on the 95% left of the data and looking at the linear fit between the original and predicted value (see Appendix A for details, Appendix A). We also ensured that the imputed length did not exceed Fishbase’s maximum reported length.

### 2.4. eDNA Metabarcoding

For each sample, environmental DNA was filtered out of 32 L of seawater in a sterile VigiDNA^®^ 0.2 µm cross-flow filtration capsule with a polyethersulfone membrane (SPYGEN, Le Bourget du Lac, France). Samples were collected as close as possible to the substrate, mostly 5 m above the seafloor. Water was pumped into the filter capsule with a disposable sterile tube connected to an Alexis^®^ peristaltic pump (Proactive Environmental Products LLC, Bradenton, FL, USA; nominal flow of 1.0 L min^−1^) and a Masterflex™ segment connected to it. On coral reefs, samples could be filtered along transects with a slow forward-going boat (~2 knots) using a reusable, extended, and weighted tube down and close to the substrate. Strict protocols were followed to avoid contamination, which included using the most disposable sterile equipment (surgical gloves, tubes, and tube joints) along with longer reusable, bleached tubes [44,55]. On continental slopes and seamount summits, four 8-L Niskin bottles (Ocean Test Equipment, Ft. Lauderdale, FL, USA) were used to collect 32 L of water at a single point for every sample. Filtration then occurred on the bridge of the ship. When the filtration process ended and all water was expelled from the filter capsules, around 80 mL of CL1 conservation buffer (SPYGEN, Le Bourget du Lac, France) was poured and enclosed in the capsule for storage and transport at room temperature. On coral reefs, eDNA transect itineraries were set to be either parallel (spread out by a few hundred meters) or following each other inside the BRUVS sample’s grid at each site. Generally, sampling of BRUVS and eDNA occurred at the same predefined coordinates.

DNA extraction was performed following an existing protocol [46]. Briefly, the DNA extraction was performed using NucleoSpin^®^ Soil (MACHEREY-NAGEL GmbH & Co., Düren, Germany), starting from step 6 and following the manufacturer’s instructions. The elution was performed by adding 100 μL of SE buffer twice. The two 50 mL tubes per filtration capsule were extracted separately, and then, the two DNA samples were pooled before the amplification step. A teleost-specific 12S mitochondrial rRNA primer pair (teleo, forward primer—ACACCGCCCGTCACTCT, reverse primer—CTTCCGGTACACTTACCATG) [44] was used for the amplification of metabarcode sequences. Because we analyzed our data using MOTUs as a proxy for species, we chose to amplify only one marker. Twelve DNA amplifications PCR per sample were performed in a final volume of 25 μL, using 3 μL of DNA extract as the template, following the protocol in [56]. The teleo primers were 5′-labeled with an eight-nucleotide tag unique to each PCR replicate with at least three differences between any pair of tags, allowing the assignment of each sequence to the corresponding sample during sequence analysis. The tags for the forward and reverse primers were identical for each PCR replicate. Negative extraction controls and negative PCR controls (ultrapure water) were amplified (with 12 replicates as well) and sequenced in parallel to the samples to monitor possible contaminations. The purified PCR products were pooled in equal volumes, to achieve a theoretical sequencing depth of 1,000,000 reads per sample. Library preparation and sequencing were performed at Fasteris (Geneva, Switzerland). A total of 18 libraries were prepared using the MetaFast protocol. A paired-end sequencing (2 × 125 bp) was carried out using an Illumina MiSeq (2 × 125 bp, Illumina, San Diego, CA, USA) using the MiSeq Flow Cell Kit v3 (Illumina, San Diego, CA, USA) or a NextSeq sequencer (2 × 125 bp, Illumina, San Diego, CA, USA) with the NextSeq Mid kit following the manufacturer’s instructions.

### 2.5. eDNA Bioinformatic

Following sequencing, reads were processed using clustering and post-clustering cleaning to remove errors and estimate the number of species using Molecular Operational Taxonomic Units (MOTUs) [57]. The methodology is described elsewhere [46]. Briefly, vsearch [58] and cutadapt [59] were used to assemble and demultiplex reads [58,59]. Swarm v.2 [60] was used to cluster sequences into MOTUs with a minimum distance of 2 mismatch between clusters. The Lower Common Ancestor (LCA) algorithm ecotag implemented in the Obitools toolkit [60] was used for taxonomic assignment of MOTUs [61] using the European Nucleotide Archive (ENA, [62]) as a reference database (release 143, March 2020). We then applied quality filters to be conservative in our estimates. To avoid spurious MOTUs originating from a PCR error, we discarded all sequences with less than 10 reads and presented only one PCR per site. Then, errors generated by tag-jumping and index-hopping [63,64] were corrected using a threshold of 0.001 of occurrence for a given MOTU within a library. Taxonomic assignments at the species level were accepted if the percentage of similarity with the reference sequence was 100%, at the genus level if the similarity was between 90 and 99%, and at the family level if the similarity was ≥85%. If these criteria were not met, the MOTU was left unassigned. The post-LCA algorithm correction threshold of 85% similarity for the family assignment was chosen to include a maximum of correct family assignments while minimizing the risk of adding wrong family identifications. Potential eDNA contamination from BRUVS bait (*Sardinops* spp.) was also removed from eDNA reads.

### 2.6. Environmental Variables

Fourteen environmental variables were identified for each sample (Appendix A). They were chosen for their potential influence on fish diversity, biomass, and assemblage structure. They included the mean, minimal, and maximal Sea Surface Temperature (SST) from NASA’s Multiscale Ultrahigh Resolution (MUR) analysis, averaged over the last 10 available years (2009–2019). The potential seafloor temperature from the Mercator global reanalysis of models constructed on satellite and in situ observations (Copernicus—CMEMS) was also used. Temperature is well known to segregate diversity at a large scale across taxa [65]. We used chlorophyll-a, suspended particulate matter, salinity, and current from the Global Ocean Satellite Observations (Copernicus—CMEMS). Chlorophyll-a concentration may indicate regions of higher energy availability and pinpoint the presence of shallow seamounts [14,66]. Suspended particulate matter concentration may differentiate oligotrophic from eutrophic nutrient zones (e.g., lagoon versus open ocean) but also seamounts, which may re-enhance nutrient internal cycling [67]. Salinity may differentiate waters closer to freshwater flux [68] and currents may influence migratory flows for species recruited on seamounts [6]. Depth was recorded for each sample as it highly structures communities, notably through light loss and associated processes [5,69]. Travel time to the nearest fish market, an index of human accessibility to natural resources, was retrieved as human pressure also impacts diversity and biomass [36,70]. The micro-habitat was also included in the BRUVS data. We evaluated, through a semi-quantitative scale [71], the distinct visually observable features (e.g., percent cover of coral, sand, vegetation, and more, see [72] for details on the method). Micro-habitat variables were used to calculate the Shannon habitat diversity index and assess whether micro-habitat diversity would influence fish diversity or biomass [73,74]. Environmental strata (“Stratum”) were also considered, as we assumed that while depth may be the main structuring variable, our environmental strata may incorporate a larger spectrum of influence that was not taken into account with the rest of the environmental variables.

### 2.7. Data Analysis

Fish species richness (BRUVS), MOTU richness (eDNA), fish abundance and biomass (BRUVS), and 14 environmental variables (Appendix A) were determined for each sample. We also computed the biomass of large predators (>50 cm carnivore or piscivore species) and sharks using our functional traits database [75]. All analyses were performed with R [76].

### 2.8. Diversity and Biomass Modelling

Boosted Regression Trees (BRTs) [49] were used to model species richness, MOTU richness, fish biomass, biomass of large predators, and shark biomass along the matrix of 14 environmental variables. The two advantages of BRTs rely on their ability to assess non-linear relationships between the response and the explanatory variables along with their ability to manage complex interactions between variables. A grid search method [49] was used to determine the best BRT hyper-parameter values (number of trees, tree complexity, learning rate, and bag fraction). BRTs with the best cross-validation (10-fold) correlation were kept and then fitted again keeping only variables with more than 5% importance in the model [49]. Cross-validation correlation was used to assess the accuracy of the models. Variable importance and marginal effects were also computed. Marginal effects allow evaluation of the “pure” effect of an explanatory variable while accounting for the effects of all other variables included in the model. Correlated explanatory variables were removed, and biomass values were transformed prior to modeling. Due to the correlated nature of micro-habitat percent cover and their poor interpretability on such a large ecological gradient (i.e., there is no coral reef at great depth due to lack of light), these were not included in the BRTs. However, habitat was included in the models with the stratum and the Shannon habitat diversity variables.

### 2.9. Comparisons across Strata

Permutational multivariate analyses (PERMANOVAs [77]) were used to compare species richness, MOTU richness, total biomass, biomass of large predators, and biomass of sharks between the five environmental strata. Significant PERMANOVAs were followed by pairwise permutation *t*-tests to identify significant factor levels. Both analyses were done with 9999 permutations. To better assess species and MOTU richness, rarefaction curves were constructed using the Hill-number method with richness as incidence data [78,79]. The Hill number’s framework considers sample size to get asymptotic richness estimates that are robust to unbalanced sampling, providing better estimation than other rarefaction methods [72,78].

### 2.10. Assemblages Structure

Principal coordinate analyses (PCoA) were used to determine the assemblages’ structure in the five environmental strata [80]. Due to heterogeneous data and the presence of double zeros in our community matrix, we used the Hellinger distance for both abundance data (BRUVS) and presence-absence data (eDNA). The Hellinger distance can accommodate heterogeneous data and allows either presence-absence or abundance data to identify communities [80]. Then, we looked at species that were common to the different environments, especially coral reef species that were also observed in at least one deep-sea environment. We looked at the functional traits of these species to determine the proportion of shared species belonging to large predators. We chose to illustrate species sharing between environments through Euler diagrams. In addition to what Venn diagrams do, Euler diagrams draw ellipses that are proportional to the defined groups of species. Shared species will prompt an intersection between ellipses that is proportional to their list size.

## 3. Results

### 3.1. Biodiversity

A total of 423 species and 791 MOTUs were recorded through 224 BRUVS and 192 eDNA samples. Boosted regression tree modeling fitted well with richness data with values of 0.89 (species richness) and 0.86 (MOTU richness) of cross-validation (CV) correlation. BRTs revealed that depth was the main driver for both species and MOTU richness, with 76.5% variable importance on BRUVS data and 74.8% on eDNA data (Table 1). The analysis of marginal effects further showed a sharp drop in richness with depth, from high richness values at shallow depth to low values at great depth (see Appendix A, Appendix A). The pattern was particularly marked for MOTU richness (Appendix A). Habitat diversity was the second most influential variable on BRUVS species richness, with 15.7% importance in the model. Accounting for the effects of other explanatory variables, the marginal effects of habitat diversity indicated higher species richness in more diverse habitats (presence of balanced, multiple feature covers). A slight interaction between depth and habitat diversity was revealed by the BRT, reinforcing the suitability of environments that combine shallow depth and high habitat diversity compared with models in which no interaction effects are allowed (Appendix A). Several other environmental variables were included in the BRT models, however, with weaker importance. Mean sea surface temperature showed 7.8% importance on species richness, with slightly more species at higher temperatures. Travel time (6.7% importance) and northward current (6.6% importance) were also retained in the MOTU richness model. Some interactions were found with these variables but remained anecdotic (Appendix A).

The major negative effect of depth on richness was reflected in the five studied strata comparison, with a steady decrease in species richness from shallow coral reefs to deep seamounts and a sharp drop of MOTU richness between coral reefs on the one hand and all deep-sea environments on the other hand (Figure 2A). PERMANOVA results on species richness revealed four significantly distinct groups: coral reefs with on average 21.8 species (±18.0 SD), followed by seamounts (50 m) with 11.7 species (±4.7 SD), then the continental slope (150 m) with 8.3 species (±4.1 SD), and finally deeper seamounts (250 and 500 m) with respectively 3.4 (±1.7 SD) and 3.0 (±1.4 SD) species per BRUVS (Figure 2A). MOTU richness was significantly higher on coral reefs (average 71.8 MOTU ± 49.7 SD) than on any deep-sea environment. Little differences were observed between seamounts (50 m) (12.9 MOTUs ± 11.0 SD), continental slopes (150 m) (10.8 MOTUs ± 9.8 SD), seamounts (250 m) (12.2 ± 14.1 SD), and seamounts (500 m) (8.3 MOTUs ± 5.2 SD) (Figure 2B).

Rarefaction curves of species and MOTU richness showed dramatically higher biodiversity on coral reefs than on any deep-sea environments, with particularly low richness on the deepest seamounts (Figure 2C,D). The pattern was especially marked for MOTU richness. Asymptotic estimates of richness were 443 species (confidence interval—CI: 403–482) and 589 MOTUs (CI: 570–620) for coral reefs, then half less species (157, CI: 119–195) and even less MOTUs (167, CI: 156–189) on seamounts (50 m), 120 species (CI: 73–168) and 189 MOTUs (CI: 168–229) on continental slopes (150 m), 37 species (CI: 25–62) and 111 MOTUs (CI: 107–122) on seamounts (250 m), and 18 species (CI: 16–27) and 74 MOTUs (CI: 70–88) on the deepest seamounts (500 m).

### 3.2. Biomass

BRT modeling of fish biomass fitted well with the data, with a cross-validation correlation of 0.73 (Table 1). Depth was again the most important explanatory variable (36.7% importance), followed by habitat diversity (23.6%), travel time (11.7%), Chla (8.1%), currents (eastward: 8.5%, northward: 5.9%), and mean SST (5.5%). The analysis of marginal effects further showed hump-shaped patterns for depth, with the highest biomass observed between approximately 50 and 300 m and the lowest biomass for shallow depth with coral reefs and the deepest strata of seamounts (500 m) (Appendix A). Habitat diversity and SST had an overall positive effect on biomass. Travel time also had a positive effect on fish biomass, with the lowest biomass values recorded near humans, and the highest in remote environments at more than 10 h travel time (Appendix A). BRT modeling of large predator biomass showed similar patterns (Appendix A). Depth was again the most important variable (26.1%), followed by habitat diversity (19.6%), travel time (15.0%), equally eastward current, Chla, and mean SST (8.7, 8.6, and 8.5%, respectively), and finally the environmental stratum (6.9%) and northward current (6.8%). Interactions found by the BRTs involved habitat diversity and northward velocity, and depth and eastward velocity with anecdotic effects (Appendix A).

Comparison of fish biomass across the five strata showed a dome-shaped pattern corresponding well to the combined effects of environmental variables retained in the BRT model, especially depth and environmental stratum (Figure 3A). PERMANOVAs further revealed that biomass levels on shallow seamounts had significantly the highest biomass (132.9 kg ± 103.4 SD), while coral reefs (54.3 kg ± 71.6 SD) showed similar biomass levels as continental slopes (150 m) (76.9 kg ± 65.7 SD) and seamounts of intermediate depth (250 m) (66.6 kg ± 63.0 SD). The deepest seamounts (500 m) showed the lowest biomass level (7.2 kg ± 5.2 SD). The pattern was identical for large predators’ biomass (Figure 3B) and sharks’ biomass (Figure 3C).

### 3.3. Assemblage Structure

Principal Coordinate Analysis showed that the five studied environments were home to relatively distinct assemblages, coral reefs, and deepest seamounts (500 m) showing the highest distinctiveness (see Appendix A, Appendix A). Coral reefs showed the second highest proportion (77%) of unique species (257 species out of 334) (Figure 4) and the highest proportion (84%) of unique MOTUs (454 MOTUs out of 540) (see Appendix A, Appendix A). Likewise, the deepest seamounts (500 m) showed the highest proportion (81%) of unique species (13 out of 16 species), although uniqueness was less with eDNA (20 out of 68 MOTUs, 29%). Other deep-sea environments showed relatively mixed assemblages, with 27% unique species (26 out of 97) and 42% unique MOTUS (62 out of 147) on seamounts (50 m), 32% unique species (23 out of 73) and 54% unique MOTUs (78 out of 144) on continental slopes (150 m), 36% unique species (9 out of 25) and 39% unique MOTUs (41 out of 106) on seamounts (250 m).

Interestingly, species that were shared between coral reefs and at least one deep-sea environment, including the deepest seamounts (black contouring in Figure 4), represented 23% of the 334 species observed on coral reefs, corresponding to 77 species. When looking at the functional traits of these coral reef species also observed in deep-sea environments, 77% (59 species) were carnivores, and 42% (32 species) were large-sized (>50 cm) carnivores, i.e., large predators (Barplot, Figure 4).

## 4. Discussion

Our study is one of the few addressing the fish biodiversity of multiple marine ecosystems using two standardized and replicated quantitative methods to provide comparative information. We showed that coral reefs may qualify as biodiversity hotspots with considerably higher species richness than any other deep-sea environments in this study. In turn, seamounts and continental slopes showed comparatively lower biodiversity both at the local scale (α-diversity) and the regional scale (γ-diversity). The deepest seamounts had on average seven times fewer species and nine times fewer MOTUs than coral reefs. Combined with the general negative effect of depth on biodiversity, our results suggest that seamounts are not hotspots for fish diversity. However, shallow seamounts surprisingly showed almost three times higher fish biomass than coral reefs, and biomass levels up to 300 m were at least equivalent. These higher biomass levels in environments between 50 and 300 m depth may represent what would be called oases for fish. Moreover, while species assemblages were distinct among the studied environments, dominant species of shallow seamounts were highly mobile large predators also observed on coral reefs, suggesting that they may use shallow seamounts as refuge from shallow coral reef anthropic pressure. Overall, our results suggest that strong conservation efforts should be prioritized on shallow seamounts and continental slopes where very high fish biomass is observed, especially for threatened large predators such as sharks. Although deeper seamounts are less rich, they are still home to unique fauna that is certainly worth protecting as well.

Our BRUVS records of coral reef fish species richness are impressively in line with previous studies around the world, with around 20–25 species average per BRUVS [81,82,83,84,85]. BRUVS have been deployed near seamounts to assess pelagic diversity [5,12], and on seamounts of abyssal depth, far outside our study’s depth range [86]. To our knowledge, this is the first study to explore a seamount’s fish biodiversity using BRUVS at summits reaching between 50 and 500 m. However, more work has been done on deep continental slopes, showing usually lower species richness on mesophotic reefs than on shallow coral reefs [74,87,88,89].

The eDNA method across studies may greatly vary depending on the volume of filtered water, the primers, the laboratory protocols, and the bioinformatic pipelines used to generate the MOTU sequences, making comparisons still binding [57,90,91]. However, the tendency to further use this method is driven by its capacity to establish better levels of species richness by integrating a larger area per sample [37,92]. On coral reefs, a study with a similar protocol highlighted more species found by eDNA than by diver-operated Underwater Visual Censuses (UVC), but with less average MOTU richness per sample than in our study (26.2 ± 12.6 SD against 71.8 ± 49.7 SD in our study) [42]. Sampling deeper strata using eDNA can also become limited when trying to assign MOTUs to referenced species, e.g., [93]. While we partly used assignment to further clean our MOTU list, we did not analyze our assigned species dataset since only 25.6% of the 791 recorded MOTU sequences were assigned to the species level.

Asymptotic estimates of total MOTU richness as well as mean MOTU richness showed unparalleled levels for coral reefs compared to all sampled deep-sea environments. This observation may be explained by the cryptobenthic fish species diversity hosted on coral reefs [94,95]. The cryptobenthic fish diversity is harder for BRUVS to capture. However, this method also showed much higher fish diversity on shallow coral reefs. BRUVS are known to sample a smaller but more representative part of the studied community compared to other methods, meaning the observed assemblages are not necessarily biased [47,48,96]. However, studies of the deep sea using video-assisted methods are bound to strong technical constraints, and our soaking times (60 min) are rarer in the literature, with longer soaking times and time-delayed videos being favored [47]. However, the involved depths in our study (around a 500-m maximum) remained relatively shallow compared to the rest of the deep-sea research that usually works at several thousands of meters.

Defining a biodiversity hotspot comes with defining threats and indices of vulnerability [17,97]. Coral reefs are acknowledged to be globally declining due to anthropic pressure [98,99]. Seamounts have been heavily fished, trawled, and exploited on a global scale [6,100]. The impacts of trawling and dredging are largely documented for seamounts and involve major erosion of biodiversity and habitat complexity [6,101,102,103,104]. Regarding our results, coral reefs may then further be praised as hotspots for biodiversity. Seamounts, on the other hand, had much lower fish species richness and therefore were not comparable to a hotspot “reference”. However, biodiversity is multifaceted, and other organisms’ richness was not studied, which could have a major impact on the overall biodiversity of seamounts. Regarding the hotspot definition applied to the marine realm, the ease of propagules spreading out in the ocean induces much more widespread species compared to land, which limits endemism in marine environments and therefore the delimitations of hotspots [17,18,19]. Further studies are certainly needed to further compare the whole biodiversity of shallow and deep-sea ecosystems. The metabarcoding of eDNA across all realms has shown promising results in that regard, e.g., [95,105].

Fish biomass followed a dome-shaped relationship with depth. Seamounts at 50 m depth had the largest biomass, followed by 150 m continental slopes and 250 m deep seamounts. Surprisingly, coral reefs showed lower biomass despite the positive effects of their habitat diversity. It is commonly accepted that biomass would decrease with respect to light, primary production, and food availability [106], and indeed, we report decreasing fish biomass between 50 and 500 m with the lowest values on the deepest seamounts. High biomass has recently been reported on continental slopes and the shelf break, e.g., [30,107]. We report even higher levels on shallow seamounts. While coral reefs are characterized by high habitat complexity, associated with important biomass [108], shallow seamounts (50 m) also harbor a strong habitat diversity, notably with extended rhodolite beds composing the substrate along with vegetation and few corals. Rhodolite beds are recognized to host high biomass as they offer substrate complexity and resources, inducing an abundance of predators [109]. Thus, our comparison of seamounts and coral reefs from the same region may promote the vision of shallow seamounts and, to a lesser extent, continental slopes as oases of biomass. Nonetheless, fishing has long exploited the resources of seamounts, and some fisheries even collapsed in the 1980s as stocks could not replenish fast enough due to the slower life cycle of deep-sea species [6,100]. Catches still have increased with time with further demand and technology to go deeper, with targeted species usually being large-bodied predators that are globally declining, e.g., [110,111,112].

This race to exploit ever deeper resources may have yet left out the shallowest seamounts, as they are not the first target of deep demersal fisheries. We showed that the assemblages of the shallow seamounts at 50 m shared many common species with coral reefs and the other deeper environments (Figure 4). A considerable proportion of these species were large predators. Deep-sea species assemblages often have a high proportion of carnivores [74,87,107], potentially caused by the shift to more heterotrophic environments [106,113]. Yet, the larger biomass of reef-associated large predators, especially sharks, on shallow seamounts should place these environments as refuges from anthropic pressure [35,114]. Seamounts are highly isolated and difficult to access features without large vessels capable of withstanding the open ocean. The travel time for the closest seamount was seven hours and up to two and a half days for the furthest, still inside the new Caledonian EEZ. We suggest that these shallow features may be of crucial ecological importance for endangered and high-value target species also found on coral reefs.

Our results on species richness are consistent across two independent sampling methods. Biomass estimated by BRUVS also seems coherent across functional traits (here size and trophic group), which are known to better reflect assemblages [115]. However, some caveats can be discussed. While we tried to use standardized methods for both shallow and deep environments, small adaptations had to be realized, such as switching between eDNA transects on coral reefs to Niskin bottles in deep environments due to technical limits. The change in protocol may have influenced the observed densities of MOTUs between coral reefs and deep environments. Currents can also increase by an order of magnitude on seamount summits. Coupled with variable degradation times and potentially lower quantities of eDNA produced by deeper species with slower metabolic rates, these factors may have also influenced the detection of MOTUs in eDNA samples [6,90,116,117]. While recent studies show remarkable site fidelity of emitted eDNA [118,119], MOTUs of species associated with the pelagic and reef environment may also not be differentiated as both environments are intimately interacting on the external barrier reef. Soaking time for BRUVS between 60 and 90 min has been estimated to provide the best samples of the shallow communities in the cost/effort ratio [47]. Due to our samples still being relatively shallow (<500 m), the demersal species assemblages remained mostly active and mobile species, with few exceptions in the deeper environments, which supports our standard soaking time of 60 min.

Furthermore, we compared the biodiversity of several environments but used only fish as an indicator. Fish diversity and biomass provide valuable indicators of ecosystem services such as regulation and linkage for ecosystem functioning (e.g., predation, consumption, sediment redistribution, nutrient recycling and redistribution, and more), food security with fisheries, and cultural services like aesthetics [120,121,122,123,124].

## 5. Conclusions

A better understanding of how biodiversity is spatially distributed is fundamental to better addressing ecosystem trajectories and issues caused by large-scale disturbances like climate change [125] or anthropic pressure [126]. The human perspective of nature is mainly utilitarian and economical, implying conservation needs to be better informed through baselines on the priorities for conservation, notably in lesser-known ecosystems like the deep sea [127]. This work aimed at helping to refine fundamental questions underlying these environments, such as seamounts, and better conceive our perception of biodiversity and its distribution across coral reefs and deeper environments. The potential of new technologies with video and eDNA metabarcoding may allow better comparative values to address biodiversity on the same baseline and compare ecosystems, regions, or habitats in order to reprioritize locations of interest for conservation and science. Our study calls for prioritizing the conservation of shallow seamounts and continental slopes since these environments support considerable fish biomass and are a refuge for large predators such as sharks, but are virtually ignored by current management plans, with only 2% of the world’s seamounts inside MPAs [5,9]. However, our study was restricted to fish, a crucial yet only small part of biodiversity. Further work looking at the whole biodiversity is warranted and may become possible with the development of key technologies such as eDNA metabarcoding.

## Figures and Tables

**Figure 1 biology-12-01446-f001:**
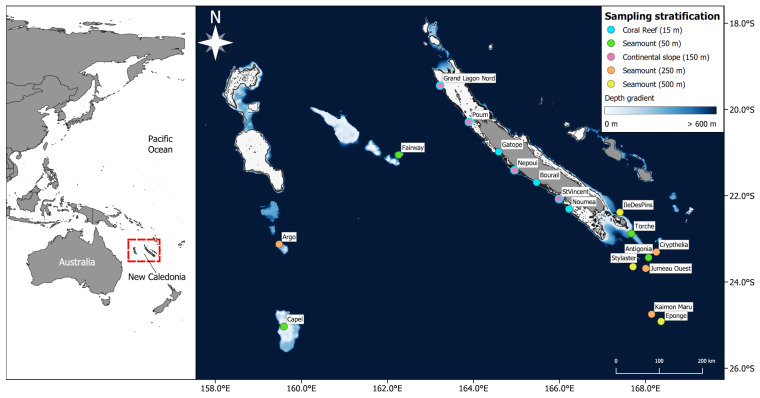
Sampling design in the five environmental strata. BRUVS and eDNA samples were collected on seven barrier coral reefs (15 m deep), four shallow seamounts (50 m summit depth), four continental slopes (150 m deep), four intermediate-depth seamounts (250 m summit depth), and three deep seamounts (500 m summit depth). Bathymetry data were derived from [50]. See Appendix A for more details on sampling design.

**Figure 2 biology-12-01446-f002:**
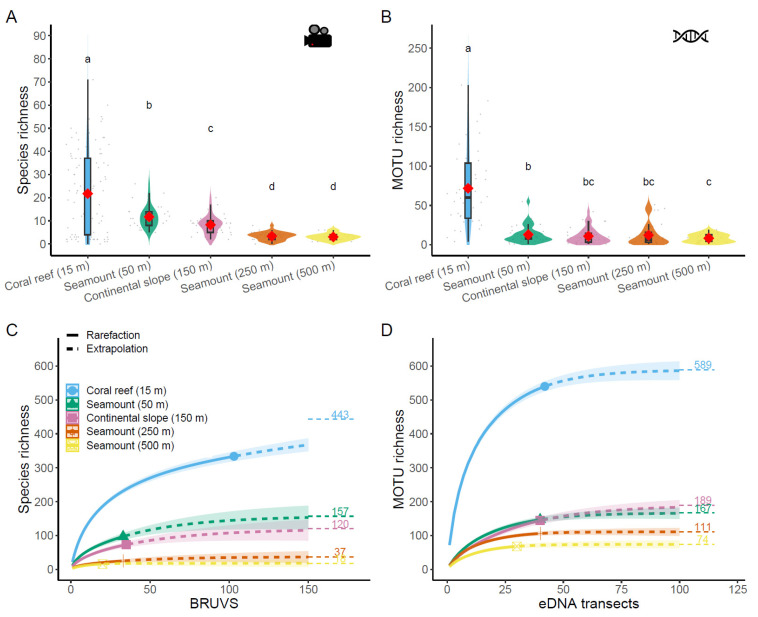
Violin plots and superimposed boxplots showing species richness observed on BRUVS (**A**) and MOTU richness in eDNA samples (**B**) for coral reef, seamounts (50, 250, 500 m), and continental slopes (150 m). The mean is represented by the red lozenge. Grey dots represent individual sample values scattered around each distribution. Significant differences at *p* < 0.05 are highlighted by grouping letters (PERMANOVAs and permutational *t*-tests with 9999 permutations). (**C**) Rarefaction curves of species richness from BRUVS and (**D**) MOTU richness from environmental DNA across coral reefs, seamounts, and continental slopes environments. The samples were rarefied (solid line) and extrapolated (dashed line) using the Hill number method [78,79]. 95% confidence intervals (CI) are shown in each respective ribbon. Horizontal lines are asymptote estimates (γ-diversity).

**Figure 3 biology-12-01446-f003:**
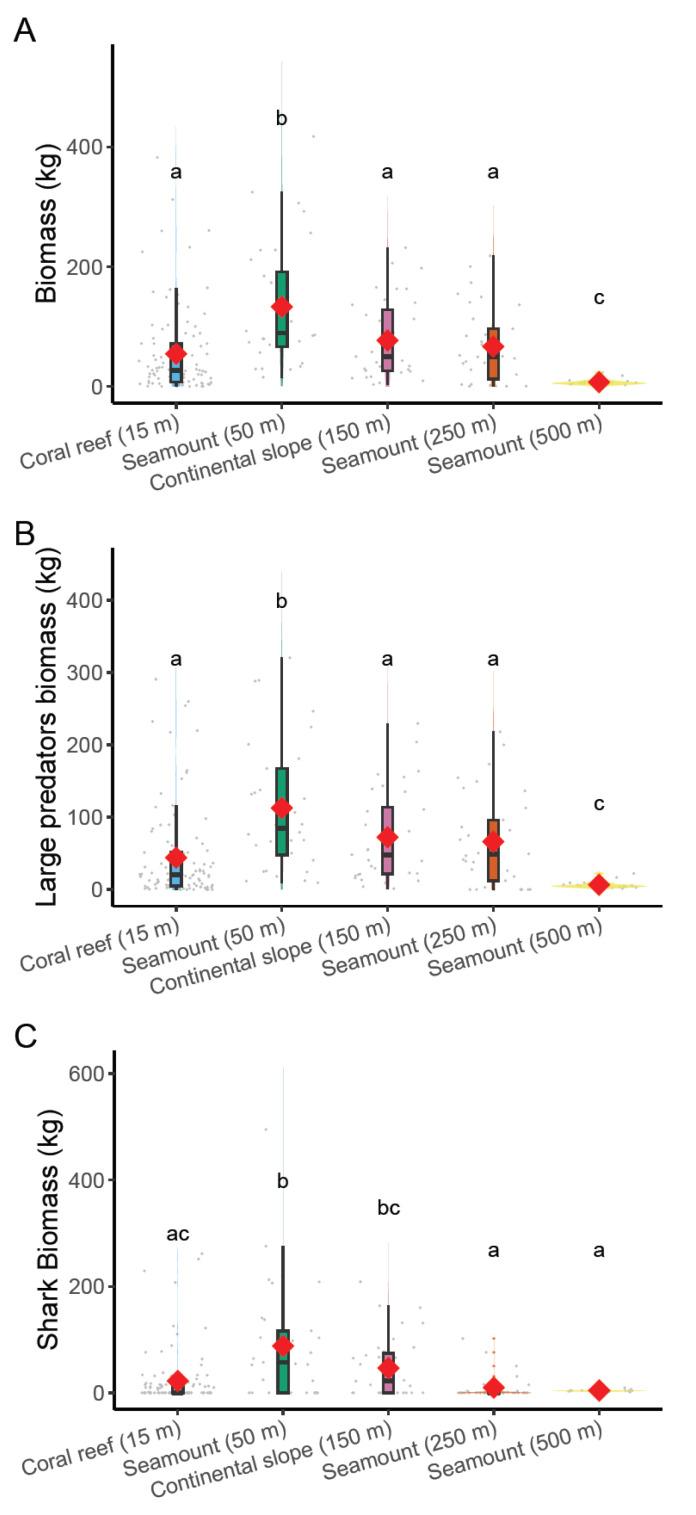
Violin plots and superimposed boxplots of biomass (BRUVS) on (**A**) all observed species, (**B**) large predators, and (**C**) sharks across 5 environmental strata: coral reefs, seamounts of variable depths (50, 250, and 500 m), and the continental slope (150 m). The mean is represented by the red lozenge. Grey dots represent individual sample values scattered around each distribution. Significant differences at *p* < 0.05 are highlighted by grouping letters (PERMANOVAs and permutational *t*-tests with 9999 permutations).

**Figure 4 biology-12-01446-f004:**
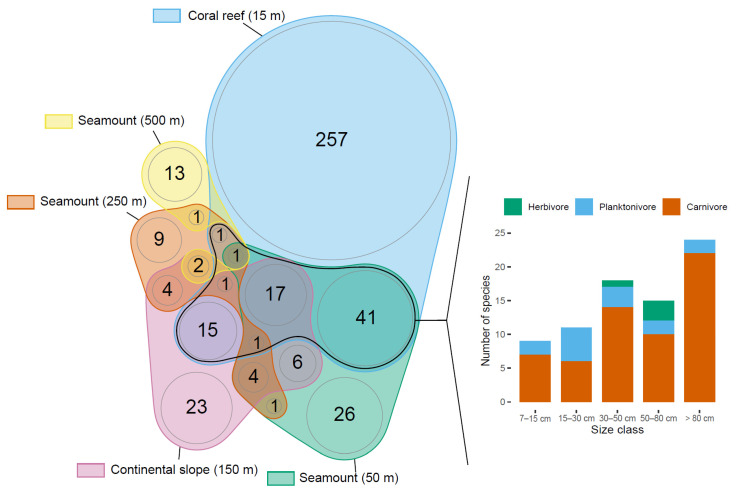
Euler diagram of species identified on BRUVS between coral reefs, seamounts of variable summit depths, and the continental slopes. Coral reef species that were also observed in at least one deep-sea environment (black grouping line) are compared in the bar plot through two functional traits: species size class and trophic group.

**Table 1 biology-12-01446-t001:** Environmental explanatory variables of several fish richness and biomass indices retained in the boosted regression trees modeling. Four different BRTs were run with species (BRUVS) and MOTU (eDNA) richness, followed by biomass (BRUVS) of full assemblages and biomass of large predators (>50 cm carnivorous and piscivorous species). Models were fit with best hyper-parameters (number of trees (NT), tree complexity (TC), learning rate (LR), bag fraction (BF) and evaluated using cross-validation correlation (CV). Explanatory variables retained in each BRT models are ordered by importance.

BRT Model	NT	TC	LR	BF	CV (SD)	Variables	Variable Importance
Species richness	1050	4	0.005	0.5	0.89	Depth	76.5%
(BRUVS)					(0.01)	Habitat diversity	15.7%
						Mean SST	7.8%
MOTU richness	700	4	0.01	0.75	0.86	Depth	74.8%
(eDNA)					(0.05)	Chla	11.9%
						Travel time	6.7%
						Northward velocity	6.6%
Total biomass	2875	5	0.001	0.75	0.73	Depth	36.7%
(BRUVS)					(0.03)	Habitat diversity	23.6%
						Travel time	11.7%
						Eastward velocity	8.5%
						Chla	8.1%
						Northward velocity	5.9%
						Mean SST	5.5%
Large predators’ biomass	2825	5	0.001	0.75	0.71	Depth	26.1%
(BRT)					(0.03)	Habitat diversity	19.6%
						Travel time	15.0%
						Eastward velocity	8.7%
						Chla	8.6%
						Mean SST	8.4%
						Environmental stratum	6.9%
						Northward velocity	6.8%

## Data Availability

The data used for this article are archived in the public repository Zenodo (https://doi.org/10.5281/zenodo.10096023 for BRUVS data, and https://doi.org/10.5281/zenodo.10095584 for eDNA data).

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
