# Peer review of "Comparing Seamounts and Coral Reefs with eDNA and BRUVS Reveals Oases and Refuges on Shallow Seamounts"

_biology, 2023, doi:10.3390/biology12111446_

Round 1

Reviewer 1 Report

Comments and Suggestions for Authors

Dear Editor,

I have reviewed the Seamounts to reef, eDNA vs BRUV MS as requested.

Apart from some more information in places to help understand things, the MS is generally a useful contribution to our understanding of patterns in coral reef fish biodiversity across major ecological biomes, as well as the comparability of eDNA datasets in this space.

My main comment is around the standardisation of habitat (or not) in the survey design given that the implication of the paper centers around "reef" habitat, yet often deeper slope or seamount habitat can be far more dominated by soft-sediment. I could not access the supplementary material that explains the micro environmental covariates used but it was good to see the proportion of reef (whether coral or rocky/sponge dominated) was recorded and used as a covariate. I appreciate habitat diversity was used as a covariate in the analysis and was a significant driver of patterns seen, in addition to depth, but it wasn't clear if this had a strong interraction with depth in the analysis due to the BRT approach not providing this level of information. Would it not be possible to use a GAM/GLM approach to this multivariate data to better tease apart the likely fact that deeper habitats are typically increasingly less coral covered or physically structured?

Some specific comments.

Para 2, while a range of things drive biodiversity, physical habitat structure, and even biogenic structure are also important. 

Para 3. Hotspots are possibly somewhat misunderstood here? A shallow system that has more species isn't necessarily hotter than a deeper one with less. Its more related to beta diversity and how that habitat (e.g. seamount) stacks up relative to similar habitats. Likewise some habitat/locations may have relatively more endemic species and be hotspots of endemism, which arguably is more of interest than simple species richness.

Last para. This is more a statement of what was done, whereas ideally it should be a more clear aim?

Methods

Lines 162-5. Needs to be clear here if the shallower seamounts were indeed isolated reefs that come up to relatively shallow depth as part of a seamount, or were just the 50 m deep extension of shallower coral reefs on the same system, and indeed if this differed again from deeper "slope" reefs

Results

I'm glad you included Fig 4 to look at the beta diversity. It does explain a lot more than a simple examination of total diversity. 

Discussion 

Line 493: The issue here isn't really if seamounts at depth are more or less "hot" than shallow areas, it is more whether they represent a different (beta) diversity. For example in a conservation planning setting you couldn't say just protect the shallow reefs that have higher alpha diversity, because all are the same and you wont add to the total diversity protected if you just add more shallow reefs. The deeper reefs have a different diversity (beta) component that is additive to shallow reefs as in Fig 4. 

Probably need some discussion on your eDNA in the sense that unlike BRUVs it isn't just sampling reef associated species but the while passing water column from the pelagic realm so likely has a lot of pelagic species as well as more of the cryptic and also non bait-attracted species. Until you can get them all to species level its not possible to separate the reef species from pelagic patterns.

Reviewer 2 Report

Comments and Suggestions for Authors

This extensive and thorough study about the biological importance of seamounts is very original and much needed. The story is generally well written and illustrated. There are, however, some major points that need attention in order to help readers understand what it is about:

-        When comparing coral reefs (15 m depth) and seamounts (50 m), one would also expect coral reefs at 50 m depth for the comparisons. Even more so than 15 m. Why is this not done?  Please explain, because now it is more like comparing apples with oranges.

-        When discussing “oasis” and “hotspot” it should be clear that a distinction is made between biodiversity and biomass (or not at all).

-        From early on it should be clear that the focus is on fish diversity / biomass.

More specific or minor points:

Line 18: ecosytem -> ecosystems (because it is one of many)

Line 21: Shallow-water coral reefs?

Line 24: deepest seamounts  -> the  deepest seamounts

Line 25:  order of magnitude” Needs to be more specific.

Line 29: call -> calls

Line 31: why does an ecosystem need to be iconic?

Line 33: at 15 m depth species coral reef assemblages are entirely different than in the mesophotic, say 50 m. It would be more logical to compare 50 m depth between fringing coral reefs and sea mounts.

Lines 33-34 and the rest of the ms: there should be a space between the numeral and the unit (15 m, 50 m, 150 m., 250 m, 500 m)

Line 36: what travel time?

Line 46: keywords should not overlap with title words

Line 52: delete “above all”. This suggests that it can be compared with the other factors, but they are very different.

Line 57: abundance distribution -> abundance patterns

Line 60-61: Delete or be more specific “with vertebrates being at the forefront”. Do you mean marine vertebrates or terrestrial ones? The latter are not relevant.

Line 62. This definition dopes not relate to ecology but to physical geography and geology.

Line 66. If the atolls are above sea-level than they are not seamounts. They are islands with some terrigenous impact and a shallow reef slope on top, also having impact on what is downslope.

Line 87: indices -> index

Lines 90-94. Oasis. This term can relate to biomass and diversity. There should be some references, eplaining both aspects. Here is a recent example from a marine biodiversity perspective: https://doi.org/10.3390/d15060779.  Later on (line 100)  the term refers to “refuge” and this is generally more related to diversity (species presence) instead of biomass.

Line 94: How can you know that there has been an increase in biomass? Why was the biomass less in the past? It would be better if the biomass is compared with that of surrounding areas.

Line 96: Replace “features” with another word, such as “landforms”. This word is too general.

Line 102: large scale disturbances -> large-scale disturbances

Line 102: Climate change is very much long-term. I suggest “the effects of climate change”

Line 106. Not all fishing is an “extreme disturbance”

Lines 114, 117. What kind of hotspots: biodiversity or biomass? Be specific because “hotspot”” by itself is not clear.

Lines 119-127. Perhaps you could mention that the main island (Grande Terre) of NC is a terrain with adjacent islands (Loyalty islands)

Line 131: seamounts fauna -> seamount fauna

Line 163: between 2m and 28m deep -> between 2 and 28 m depth

Line 164: between 45m and 570m -> between 45 and 570 m

Lines 166-169. There should not be a digit behand the decimal mark. Measurements were not that precies.

Lines 155 and on. Numerals < 10 not combined with metrical units should be written as words: seven barrier reefs, four seamounts, etc. But “5-16 per site” is ok, although it should be “5–16 per site” with an n-dash instead of a hyphen.

Line 117: 300-500 meter -> 300–500 m

Line 178. Individuals of what? Fish?

Line 181: 1.5-meter -> 1.5-m

Line 182: The “spp.” in “Sardinops spp.” should not be in italic script

Line 185: 1920x1080 pixel -> 1920×1080-pixel

Line 189: “standard” and “most used” Both are the same to me. Select one.

Lines 198-190. What relationship from Fishbase?

Line 200. Unclear what is meant by “length structure” and “general pattern”.  Please rephrase.

Line 202: up to ten -> up to 10

Line 206: “Latitude and longitude of …. lengths”. Do you mean the coordinates of the localities where length records were taken?

Line 211: max -> maximum

Line 218: 5 meters -> 5 m / five meters

Line 227: 32 liters -> 32 L

Line 239: two 50-mL tube -> two 50-mL tubes

Line 240: 15,000×g -> 15,000 g ?

Line 284: a taxonomy -> taxonomic identification / taxonomic classification ?

Line 286: same. Taxonomy is a scientific discipline.

Line 298: wrong family assignments in the family detections -> wrong family identifications ?

Line 299. “spp.” not italic

Line 305: ten -> 10

Line 317. Other factors: water pressure, temperature ?

Line 317. Travel time from where and by whom? Do you mean accessibility by humans?

Line 351. Anderson 2001 requires a reference number?

Line 353: 5 -> five

Line 356, 357: Hill number method -> Hill-number method

Figure 2: there should be a space between the numeral and the unit (50 m, 150 m., 250 m, 500 m); “deep slope” should be “continental slope” as seamounts also have deep slopes.

Line 427: PERMANOVAS -> PERMANOVAs

Figure3: See remark Fig 2. Kg -> kg

Line 461: PERMANOVAS -> PERMANOVAs

Figure 4: See remark Fig 2.

Line 492: 7 times -> seven times; 9 times -> nine times

Lines 505: 20-25 -> 20–25

Lines 506, 506. Delete “e.g.,”. Redundant.

Line 509: meters -> m

Line 526. Cryptobenthic diversity of fish, fauna, biota ? Please, specify.

Line 571. Unclear what is meant by “to hog” in the present context.

Line 573: at 50 meters -> at 50 m depth

Line 581: new caledonian -> New Caledonian

Line 604. Aestethics. Not many people have the chance to enjoy what occurs at depths > 50 m.

Line 606. Better understanding how biodiversity -> A better understanding of how biodiversity

Lines 621-648. There is much explanation about the supplementary materials. This is not very useful. It would be more helpful if the explanations are in the supplementary files themselves

Reviewer 3 Report

Comments and Suggestions for Authors

Study design and research questions are clearly described. In this sense, it is easy to understand the aim of this study. The bright side of the manuscript is that to provide practical details and results in related topic. In this context, the study contributes to understanding biodiversity of shallow seamounts. Only minor concerns are raised. Therefore, I would like to make some suggestions to improve the quality of the paper as below:

Lines 44-45: “They call for prioritizing the conservation of shallowest seamounts that form biomass oases and refuges for threatened megafauna.” Please rephrase the sentence and emphasise the major contribution of the study.

Lines 51-53: “Often considered as biodiversity hotspots [2], oases [3,4] and refuges for marine megafauna [5], seamounts are above all the least studied of the major biomes of the global ocean.” I think such sentence or similar sentence would better fit here -> “Although seamounts often considered as biodiversity hotspots [2], oases [3,4] and refuges for marine megafauna [5], they are the least studied landforms of the major biomes of the global ocean.”

Line 53: “less than 0.002%” -> “only less than 0.002%”

Lines 75-76: “With an enhanced biodiversity compared to their surrounding environments, seamounts have long been considered as hotspots, although the claim is now largely debated” Please rephrase the sentence.

Lines 88-90: “Biomass seems to share synergistic links with biodiversity, but climate, biological production and yet unknown constraints also enhance biomass [21].” Please rephrase the sentences and explain the biomass-biodiversity relationship with 1-2 sentences.

Line 174: I could not find the refence “Shom-IRD (2021)”.

Line 194: Please explain how species were identified with 1-2 sentences and a reference.

Line 403: “Permanovas” -> “PERMANOVA results”

Line 605: Please explain the limitations of the study and future remarks. 
